# Enhanced Radar Detection in the Presence of Specular Reflection Using a Single Transmitting Antenna and Three Receiving Antennas

**Yong Yang * and Xue-Song Wang**

School of Electronic Science, National University of Defense Technology, Changsha 410073, China
* Correspondence: youngtfvc@163.com

**Abstract:** Radar target echoes undergo fading in the presence of specular reflection, which is adverse to radar detection. To address this problem, this paper proposes a radar detection method that uses a single transmitting antenna and three receiving antennas. The proposed method uses the maximum absolute value of the difference in the radar received signal power among the three receiving antennas as the test statistic. First, the target echo in the presence of specular reflection is analyzed. Then, selection of the required number of radar antennas and the heights at which they must be situated are discussed. Subsequently, analytical expressions of the radar detection probability and the false alarm probability are derived. Finally, simulation results are presented, which show that the proposed method improves radar detection performance in the presence of specular reflection.

**Keywords:** radar detection; specular reflection; multiple antennas; detection probability

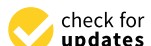



## 1. Introduction

Multipath interference affects the radar detection of low altitude targets over the sea. In multipaths, the directly arriving target echo and the sea surface-reflected target echo overlap with each other. The resulting combined multipath returns combine either constructively or destructively with the direct-path signal, which produces a stronger or weaker total received signal at random. In most cases, multipath attenuates the radar received signal intensity. It is disadvantageous to radar detection [1–5]. Therefore, overcoming the negative effect of multipath is a key issue for radar detection of low altitude targets over the sea.

Multipath scattering from a rough surface involves two components: specular reflection and diffuse reflection. In most cases, specular reflection dominates multipath scattering, so many scholars only consider specular reflection for multipath scattering [6,7]. In the specular reflection case, the radar detection performance can be improved if prior knowledge of the radar-target environment is known [8]. However, the target position usually varies with time, and such prior knowledge is difficult to obtain. Without prior knowledge, radar systems often employ frequency diversity to overcome multipath interference [9–11]. For example, [10] developed adaptive orthogonal frequency-division multiplexing (OFDM) signals for moving target detection in multipaths. In [11], an order statistics-based detection method was proposed to improve the radar target detection performance in multipaths based on frequency diversity. In fact, multipath returns have different propagation ways. This implies that the amplitude or the phase of the multipath returns are different at some fixed location in the space. Thus, spatial diversity can be used to overcome the negative effect the multipath [12,13]. For example, MIMO radar is proposed to detect the target in the presence of multipath [14]. However, MIMO radar is difficult to be implemented in reality. In addition, array antennas and multiple subapertures (which are collectively referred to as multiple antennas) are widely used for radar tracking in multipaths [15–18]. Since multiple antennas can be used for radar tracking in this way, it is likely that they can also be used for radar detection in multipaths. However, to our knowledge, there has been

little previous work on radar detection in multipaths using multiple antennas. Whether radar detection performance in multipaths can be enhanced by using multiple antennas is an interesting question. There are some problems that need to be addressed when using multiple antennas for radar detection in multipaths:

(1)   How many antennas should be used in the system?
(2)   Where should the antenna heights be set?
(3)   What test statistic should be used?

In this paper, we propose using a single transmitting antenna and three receiving antennas to improve radar detection performance in the presence of specular reflection. The maximum absolute value of the difference in the radar-received signal power among the three antennas is used as the test statistic. The advantages of using three receiving antennas are demonstrated, and the requirements for setting the radar antenna heights are discussed. Mathematical expressions for the radar detection probability and the false alarm probability are derived. Simulation results are given to validate the proposed method.

The rest of this paper is organized as follows. Radar target echoes in the presence of specular reflection are analyzed in Section 2. In Section 3, the selection of the number of antennas and the antenna heights are discussed. Mathematical derivations of the radar detection probability and the false alarm probability are presented in Section 4. The simulation results that demonstrate the validity of the proposed method are given in Section 5. Finally, Section 6 concludes the paper.

## 2. Radar Target Echo in the Presence of Specular Reflection

Assuming that radar transmits and receives signals using the same single antenna, a schematic diagram of the radar specular reflection is shown in Figure 1. In the presence of specular reflection, four different paths contribute to the radar-received target echoes: direct-direct (ABA), direct-reflected (ABOA), reflected-direct (AOBA) and reflected-reflected (AOBOA) paths [19]. Then, the received target echoes can be expressed by

$$
s_t = A \exp(\mathrm{j}\varphi) \left[ 1 + \sqrt{\frac{G_r(\theta_r)}{G_r(\theta_d)}} \rho_s \exp(\mathrm{j}\phi) + \sqrt{\frac{G_t(\theta_r)}{G_t(\theta_d)}} \rho_s \exp(\mathrm{j}\phi) + \sqrt{\frac{G_t(\theta_r) G_r(\theta_r)}{G_t(\theta_d) G_r(\theta_d)}} \rho_s^2 \exp(\mathrm{j}2\phi) \right],
\tag{1}
$$

where $A$ and $\varphi$ are the amplitude and phase of the directly arriving target echo, respectively; $G_t(\theta)$ and $G_r(\theta)$ are the radar transmitting and receiving antenna gains at an angle $\theta$, respectively; $\theta_d$ and $\theta_r$ are the elevation angles of the direct and reflected paths, respectively; $\rho_s$ is the amplitude of the specular reflection coefficient; $\phi = \phi_l + \phi_\rho$, where $\phi_\rho$ is the phase of the specular reflection coefficient, which is a constant; $\phi_l = \frac{2\pi}{\lambda}(l_r + l_t - R)$, where $\lambda$ is the wavelength; and $R$, $l_r$, and $l_t$ are the lengths of paths AB, AO, and BO, respectively, as shown in Figure 1. Clearly, $\phi$ and $\phi_l$ are both sensitive to the radar height $h_r$ and to the target height $h_t$.

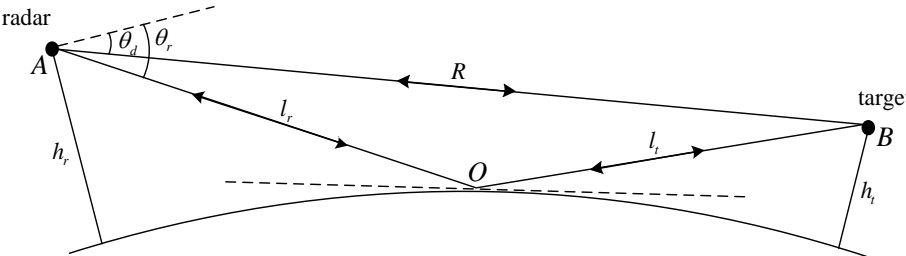

**Figure 1.** Schematic diagram of radar specular reflection.

In the case when the radar operates at low grazing angles, $\theta_r \approx \theta_d$, $G_t(\theta_r) \approx G_t(\theta_d)$, and $G_r(\theta_r) \approx G_r(\theta_d)$. Thus, Equation (1) can be simplified to

$$s_t = A \exp(\mathrm{j}\varphi)[1 + \rho_s \exp(\mathrm{j}\phi)]^2, \tag{2}$$

From (2), it can be seen that the target echo power varies as the $\phi$ varies. The $\phi$ varies with changes in the radar antenna height, target height and target distance. Therefore, the target echo power varies with changes in the radar antenna height and the target location. To demonstrate this conclusion indirectly, Figure 2 presents $\phi_l$ under various radar antenna heights and target locations, where the radar wavelength is 0.03 m.

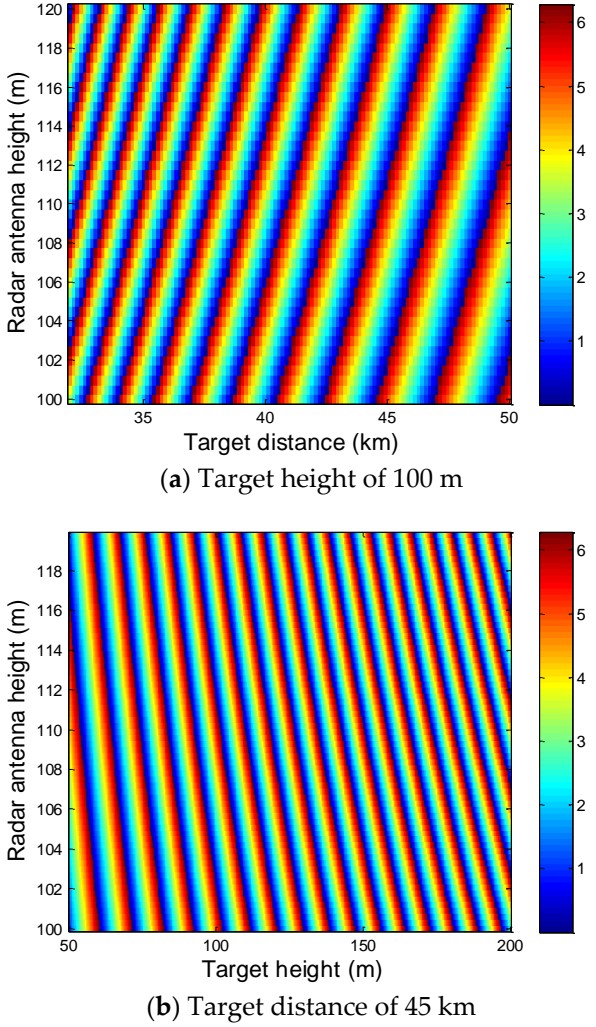

(**a**) Target height of 100 m

(**b**) Target distance of 45 km

**Figure 2.** $\phi_l$ versus radar antenna height and target location.

In Figure 2, $\phi_l$ varies with changes in the radar antenna height, target height and target distance. Therefore, $\phi$ also varies with the radar antenna height, target height and target distance, which induces the target echo power to vary in the same manner. Thus, we can conclude that the target echo powers that are received by antennas at different heights are also likely different. However, the clutter mean powers that are received by the antennas at the different heights are almost identical if the height difference between antennas is within several meters. Because the ground surface or sea surface represents an area target, the clutter mean power is mainly related to the distance between the radar antenna and the reflected surface. The height differences between the multiple antennas are much smaller than the distance between the radar and the reflected surface, which will not induce the clutter mean power difference in antennas. Overall, the target echo powers received by

each of the multiple antennas may be different, whereas the clutter mean powers received by the same multiple antennas are almost identical. Thus, we can use multipath antennas at different heights and use the differences in the signal powers received by the multiple antennas as test statistics to decide whether a target exists or not. In the following section, we first discuss how to select the antenna number and their heights.

### 3. Selection of Antenna Numbers and Heights

Assuming that the radar uses a single transmitting antenna and multiple receiving antennas and that these antennas are set at different heights. To better detect the target, the differences among the target echo powers received by the multiple antennas needs to be obvious at all times. However, the target location is unknown beforehand, and this location changes with time, which may cause the differences among the target echo powers received by the multiple antennas to be obvious when the target is at certain locations and small when the target is at other locations. Therefore, selection of the number of antennas and their heights is vital to ensure that there is at least a difference among the target echo powers received by the multiple antennas, which is always obvious for any target location.

#### 3.1. Selection of Antenna Number

Assuming that the radar uses a single transmitting antenna and two receiving antennas. Antenna one both transmits and receives signals, while antenna two only receives signals. The simulated target echo powers that are received by the two antennas are shown in Figure 3, where "ant 1" and "ant 2" denote antenna one and antenna two, respectively. In the simulations, the two antenna heights are randomly set at 200 m and 210 m. The radar transmitted power is 50 kW, the radar wavelength is 0.03 m, the maximum antenna gain is 43 dB, the half power beam width is 4°, and the target height is 50 m. The target maneuver is not considered in this paper. Please refer to [20] for maneuvering target detection.

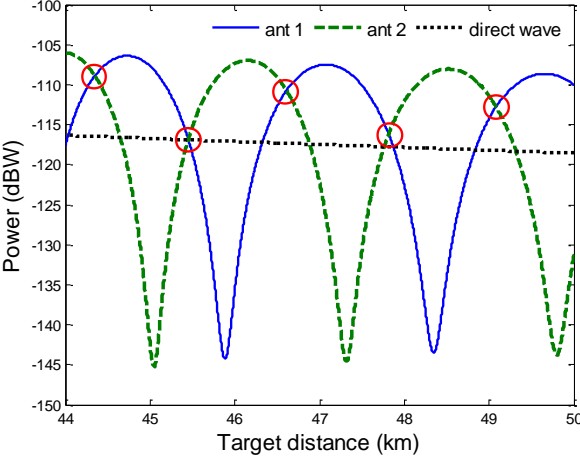

**Figure 3.** Received target echo powers received by two antennas with different heights.

Figure 3 shows that the target echo powers that are received by the two antennas are obviously different in most cases. However, regardless of the height difference between the two antennas, there are always some target locations at which the difference between the received target echo powers of the two antennas is small, such as the locations that are labeled with circles in Figure 3. The radar detection probability will be low at these locations if we use the difference between the received target echo powers of the two antennas as the test statistic. To address this problem, we attempt to add one receiving antenna and use a single transmitting antenna and three receiving antennas.

Denoting the target echo powers received by the three antennas as $z_1$, $z_2$ and $z_3$. The differences between the target echo powers are $|z_1 - z_2|$, $|z_1 - z_3|$ and $|z_2 - z_3|$. Then, we choose the maximum value of the above differences as the test statistic. Thus, while

$|z_1 - z_2|$ may be small at certain locations, either $|z_1 - z_3|$ or $|z_2 - z_3|$ may be apparent at these locations. To demonstrate this prediction, Figure 4 presents the simulated target echo powers that were received by three antennas set at different heights, where "ant 3" denotes antenna three, which only receives signals, and the heights of the three antennas are set at 200 m, 206 m and 212 m. The other parameters are the same as those used in Figure 3.

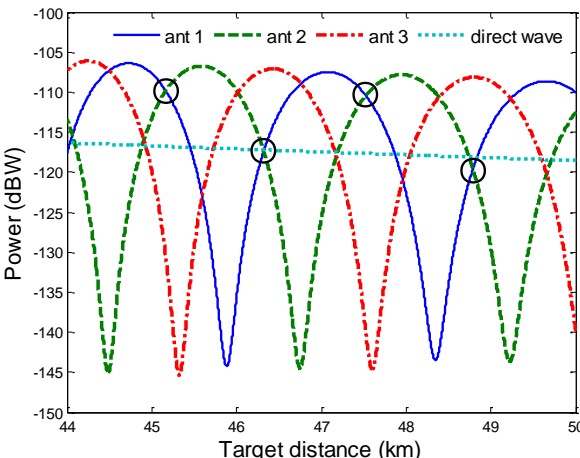

**Figure 4.** Target echo powers received by three antennas with different heights.

Figure 4 shows that the differences among the target echo powers that are received by the three antennas are apparent in most cases. Despite the fact that $|z_1 - z_2|$ is small in the locations labeled by circles, $|z_1 - z_3|$ or $|z_2 - z_3|$ is apparent at these locations. This finding verifies our prediction above and illustrates the value of using the third antenna. Next, we will discuss how to set the heights of the three antennas.

### 3.2. Setting of Antenna Heights

From Figure 2, it can be seen that the target echo power is sensitive to the radar antenna height. In addition, Figures 3 and 4 show that the differences among the target echo powers of the multiple antennas are related to the antenna heights. Therefore, setting the antenna height carefully is important to ensure that there is always an apparent difference in the target echo powers of multiple antennas for any target location. In general, there are four antenna height setting requirements:

(1)    The antenna height must be set high enough to ensure that the target is within the radar line of sight region when considering the earth's curvature.

(2)    The distance between the transmitting and receiving antennas should be sufficient to maintain a fixed antenna isolation degree.

(3)    The clutter mean powers received by the three antennas are approximately equal to each other.

(4)    There is always an apparent difference value for the target echo powers of the multiple antennas for any target location.

To meet the first requirement, the minimum radar antenna height can be calculated by referring to the work of [21]. The second and third requirements are also both easily satisfied if the distances between the three antenna heights are moderate. The fourth requirement is the most important and is the most difficult to satisfy. As Figures 2 and 3 indicate, the target echo power varies with changes in the target height and the target distance, regardless of the antenna height. Thus, the maximum target echo power difference for the three antennas varies with changes in target location, and it may be small or apparent as the target location changes. For this reason, there is no optimal height setting available for the three antennas that will satisfy the fourth requirement. Without loss of generality, we set the three antenna heights to 200 m, 206 m and 212 m. A schematic diagram of the configuration of the three antennas is shown in Figure 5. Because the three antennas are

identical, it can be assumed that antenna 1 randomly transmits and receives signals, while antenna 2 and antenna 3 receive signals only. In the next step, the detection probability and the false alarm probability for the radar using the three antennas will be derived.

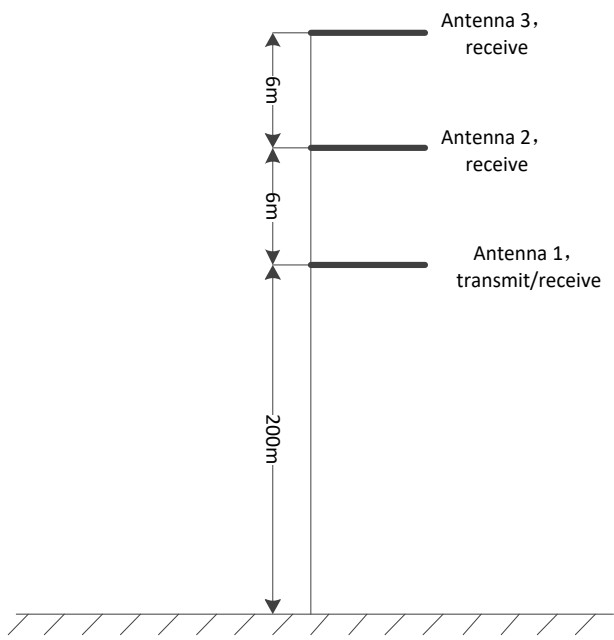

**Figure 5.** Schematic diagram of the configuration of the three antennas.

## 4. Detection and False Alarm Probabilities

In the presence of specular reflection, the signal received by the $i$th antenna can be written as

$$x_i = \begin{cases} c_i + n_i & , \ H_0 \\ s_i[1 + \rho_s \exp(j\phi_i)]^2 + c_i + n_i & , \ H_1 \end{cases} \quad i = 1,2,3, \tag{3}$$

where $H_0$ and $H_1$ denote that the target is absent and present, respectively, the subscript $i$ denotes the $i$th antenna, $s_i$ denotes the directly arriving target echo, $c_i$ represents the complex Gaussian distributed clutter with zero mean and variance $\sigma_c^2$, $n_i$ is the complex Gaussian distributed thermal noise with zero mean and variance $\sigma_n^2$, and the clutter and the thermal noise are independent of each other.

Expanding (3) gives the received signal of the $i$th antenna under $H_1$ as

$$\begin{aligned} x_i &= s_i[1 + \rho_s \exp(j\phi_i)]^2 + c_i + n_i \\ &= s_x a_x - a_y s_y + c_x + n_x + j(a_y s_x + a_x s_y + c_y + n_y) \end{aligned}, \tag{4}$$

where $a_x = 1 + 2\rho_s \cos\phi + \rho_s^2 \cos 2\phi$, $a_y = \rho_s^2 \sin 2\phi + 2\rho_s \sin\phi$, $s_i = s_x + js_y$, $c_i = c_x + jc_y$, and $n_i = n_x + jn_y$.

For the Swerling I fluctuation target, its mean power is denoted by $P_s$. Then, based on (4), the real and imaginary parts of $x_i$ can be derived, and both are found to have Gaussian distributions with zero means and variances in $\left(a_x^2 + a_y^2\right)P_s/2 + \sigma_c^2 + \sigma_n^2$. Because $z_i = |x_i|^2$, the probability density function (PDF) of $z_i$ under $H_1$ can be obtained as

$$\begin{aligned} f(z_i|H_1) &= \frac{1}{\left(a_x^2 + a_y^2\right)P_s + P_n + P_c} \cdot \\ &\quad \exp\left[-\frac{z_i}{\left(a_x^2 + a_y^2\right)P_s + P_n + P_c}\right], i = 1,2,3 \end{aligned}, \tag{5}$$

where $P_n = 2\sigma_n^2$ and $P_c = 2\sigma_c^2$.

From (5), the PDF of $z_{12} = |z_1 - z_2|$ under $H_1$ can be derived by [22]

$$
\begin{aligned}
f(z_{12}|H_1) &= \int_0^\infty f_{z_1}(z_{12} + z_2|H_1)f(z_2|H_1)\mathrm{d}z_2 + \\
&\quad \int_{z_{12}}^\infty f_{z_1}(z_2 - z_{12}|H_1)f(z_2|H_1)\mathrm{d}z_2 \\
&= \frac{1}{\left(a_x^2 + a_y^2\right)P_s + P_n + P_c} \cdot \\
&\quad \exp\left[-\frac{z_{12}}{\left(a_x^2 + a_y^2\right)P_s + P_n + P_c}\right]
\end{aligned}
\tag{6}
$$

where $f_{z_1}(z_{12} + z_2|H_1)$ means the PDF of $z_1$ under $H_1$ with independent variable $z_1$ substituted by $z_{12} + z_2$.

Similarly, the probability density functions of $z_{13}$ and $z_{23}$ are the same as that of $z_{12}$. Choosing the test statistic as

$$
L = \max(z_{12}, z_{13}, z_{23}),
\tag{7}
$$

The radar detection probability can be calculated by [see the Appendix A]

$$
\begin{aligned}
P_d &= \Pr[\max(z_{12}, z_{13}, z_{23}) > \eta|H_1] \\
&= 1 - \Pr[|z_1 - z_2| < \eta \cap |z_1 - z_3| < \eta \cap |z_2 - z_3| < \eta] \\
&= 1 - \exp\left(-\frac{3\eta}{\chi}\right)\left[\exp\left(\frac{\eta}{\chi}\right) - 1\right]^3,
\end{aligned}
\tag{8}
$$

where $\eta$ is the detection threshold, and $\chi = \left(a_x^2 + a_y^2\right)P_s + P_n + P_c$.

Setting $P_s = 0$ in (8) then yields the radar false alarm probability as

$$
P_f = 1 - \exp\left(-\frac{3\eta}{P_n + P_c}\right)\left[\exp\left(\frac{\eta}{P_n + P_c}\right) - 1\right]^3,
\tag{9}
$$

## 5. Simulation Results and Analysis

In this section, the detection performance of the radar using a single transmitting antenna and three receiving antennas in the presence of specular reflection is presented by simulation. We compare the detection performance with that of the radar employing a single antenna and that of the radar employing a single transmitting antenna and two receiving antennas.

For the radar employing a single transmitting antenna and two receiving antennas, the test statistic is $z_{12}$, and the radar detection probability is given by

$$
P_d = \int_\eta^\infty f(z_{12})\mathrm{d}z_{12} = \exp\left[-\frac{\eta}{\left(a_x^2 + a_y^2\right)P_s + P_n + P_c}\right],
\tag{10}
$$

By letting $P_s = 0$ in (10), we acquire the false alarm probability for the radar employing a single transmitting antenna and two receiving antennas as

$$
P_f = \exp\left(-\frac{\eta}{P_n + P_c}\right),
\tag{11}
$$

The detection probability and the false alarm probability for the radar using a single antenna are the same as (10) and (11), respectively. Therefore, in the presence of specular reflection, the detection performance of the radar using a single transmitting antenna and two receiving antennas is the same as that of the single antenna radar.

In the simulations, the clutter mean power $P_c = 10$, the thermal noise mean power $P_n = 1$, and $\rho_s = 0.9$. Figure 6 shows the false alarm probability versus detection threshold, where the notation "three antennas" means radar employing a single transmitting antenna and three receiving antennas, and the notation "single antenna" denotes the single antenna radar.

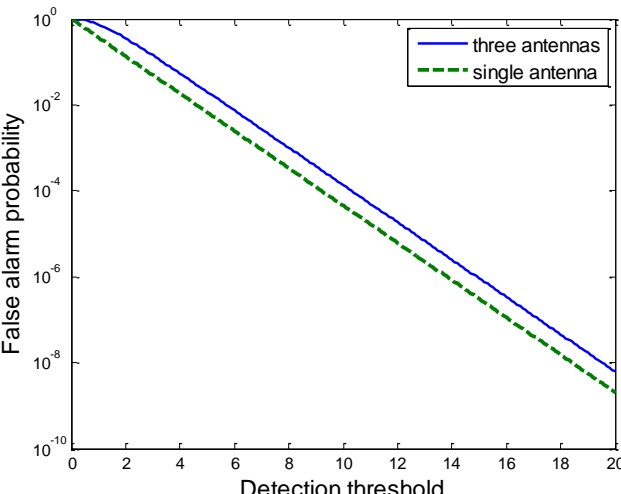

**Figure 6.** False alarm probability versus detection threshold.

The detection threshold can be obtained by interpretation according to Figure 6 when the false alarm probability is fixed. Figure 6 illustrates that the detection threshold for the radar with the single transmitting antenna and three receiving antennas is higher than that for the single antenna radar to maintain a constant false alarm probability.

The theoretical and simulated detection probabilities for the radar employing a single transmitting antenna and three receiving antennas in the presence of specular reflection are shown in Figure 7, where the Monte Carlo simulation times are 10,000, $P_f = 10^{-3}$, and $\phi_l = \pi$. In addition, the corresponding detection probabilities of the radar using a single antenna in the presence and absence of specular reflection are also presented for comparison. In Figure 7, the notations "with" and "without" denote the presence and absence of specular reflection, respectively.

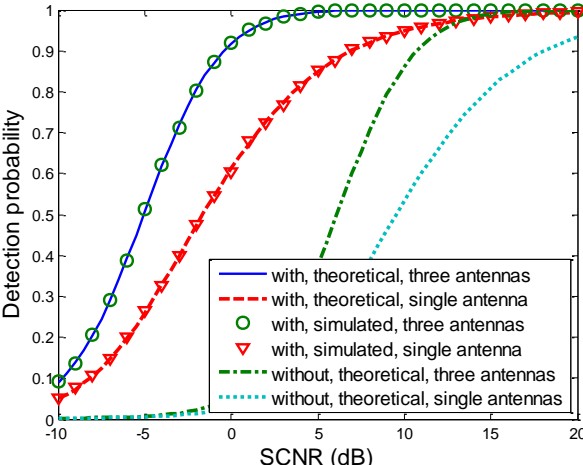

**Figure 7.** Theoretical and simulated detection probabilities for radar using a single transmitting antenna and three receiving antennas.

Figure 7 shows that the simulated radar detection probability agrees well with the theoretical radar detection probability, which demonstrates the correctness of the theoretical derivation in Section 4. In addition, Figure 7 illustrates that the detection performance of the radar using a single transmitting antenna and three receiving antennas is better than that of the radar with a single antenna.

Because the target echo power is sensitive to $\phi_l$ and $\phi_l$ varies with changes in the target location, we present the radar detection probabilities under various $\phi_l$ of the first antenna in Figure 8. The corresponding detection probabilities of the single antenna radar under

the same $\phi_l$ values are presented in Figure 8 for comparison. Figure 8 shows that radar detection performance in the presence of specular reflection varies with respect to changes in $\phi_l$. However, the detection probability for the radar with a single transmitting antenna and three receiving antennas is always higher than that for the single antenna radar, which verifies the effectiveness of the proposed method.

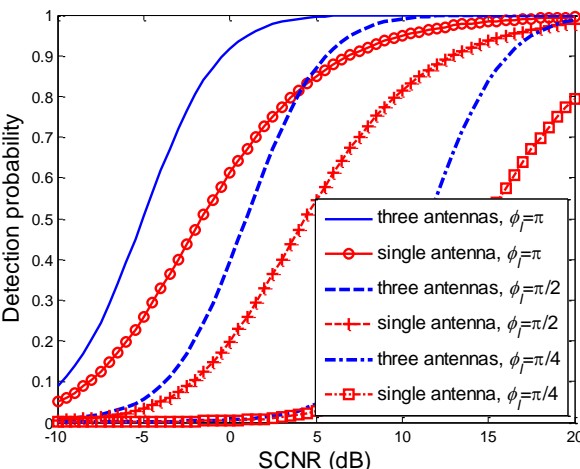

**Figure 8.** Radar detection probabilities under various $\phi_l$.

## 6. Conclusions

In this paper, radar utilizing a single transmitting antenna and three receiving antennas is proposed for target detection in the presence of specular reflection. This method takes advantage of space diversity to overcome the passive effects of specular reflection on radar detection performance. Based on the characteristic that the target echo powers for antennas set at different heights in the presence of specular reflection are different, the method takes the maximum absolute value of the differences in the received signal powers among the three receiving antennas as the test statistic. Analytical expressions of the radar detection probability and the false alarm probability are obtained. Simulation results show that radar detection performance in the presence of specular reflection is enhanced when using the proposed method.

The impact of the number of receiving antenna is not analyzed concretely in this paper. How to choose the number of the receiving antenna and how to set their height need further investigation. In addition, validation of the proposed method using experimental data is also needed to be studied in the future.

**Author Contributions:** Conceptualization, Y.Y. and X.-S.W.; methodology, Y.Y. and X.-S.W.; software, Y.Y.; validation, Y.Y. and X.-S.W.; writing—review and editing, Y.Y.; project administration, Y.Y.; funding acquisition, Y.Y. All authors have read and agreed to the published version of the manuscript.

**Funding:** This work is supported by the National Natural Science Foundation of China under Grant 62171447.

**Data Availability Statement:** Data sharing is not applicable to this article.

**Conflicts of Interest:** The authors declare no conflict of interest.

**Appendix A**

Because $z_{12}$, $z_{13}$ and $z_{23}$ are similar to each other, we have

$$
\begin{aligned}
\mathrm{Pr} \quad & [\max(z_{12}, z_{13}, z_{23}) < \eta] \\
= \quad & \mathrm{Pr}[z_{12} < \eta, z_{12} > z_{13}, z_{12} > z_{23}] + \\
& \mathrm{Pr}[z_{13} < \eta, z_{13} > z_{12}, z_{13} > z_{23}] + \\
& \mathrm{Pr}[z_{23} < \eta, z_{23} > z_{12}, z_{23} > z_{13}] \\
= \quad & 3 \cdot \mathrm{Pr}[z_{12} < \eta, z_{12} > z_{13}, z_{12} > z_{23}] \\
= \quad & 3 \cdot \mathrm{Pr}[z_{12} < \eta, z_{12} > \max(z_{13}, z_{23})]
\end{aligned}
\tag{A1}
$$

Using the notation $t = \max(z_{13}, z_{23})$, (A1) can then be given by

$$
\mathrm{Pr}[\max(z_{12}, z_{13}, z_{23}) < \eta] = 3 \int_0^\eta \int_t^\eta f(z_{12}) \mathrm{d}z_{12} \cdot f(t) \mathrm{d}t,
\tag{A2}
$$

where $\eta$ is the detection threshold, and

$$
f(t) = F_{z_{13}}(t) f_{z_{23}}(t) + f_{z_{13}}(t) F_{z_{23}}(t),
\tag{A3}
$$

where $f_{z_{23}}(t) = f_{z_{13}}(t) = \frac{1}{\chi} \exp\left[-\frac{t}{\chi}\right]$, $\chi = \left(a_x^2 + a_y^2\right) P_s + P_n + P_c$, and $F_{z_{13}}(t) = F_{z_{23}}(t) = \mathrm{Pr}[z_{23} < t] = \int_0^t f(z_{23}) \mathrm{d}z_{23} = 1 - \exp\left[-\frac{t}{\chi}\right]$. Then, $f(t)$ can be further simplified to

$$
f(t) = 2 F_{z_{13}}(t) f_{z_{23}}(t) = \frac{2}{\chi} \exp\left(-\frac{t}{\chi}\right) - \frac{2}{\chi} \exp\left(-\frac{2t}{\chi}\right),
\tag{A4}
$$

Substituting (A4) and (6) into (A2) yields

$$
\mathrm{Pr}[\max(z_{12}, z_{13}, z_{23}) < \eta] = \exp\left(-\frac{3\eta}{\chi}\right) \left[\exp\left(\frac{\eta}{\chi}\right) - 1\right]^3,
\tag{A5}
$$

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
