# Peer review of "Enhanced Radar Detection in the Presence of Specular Reflection Using a Single Transmitting Antenna and Three Receiving Antennas"

_remotesensing, doi:10.3390/rs15123204_

Round 1
Reviewer 1 Report
1- The paper is mostly similar with the one published 2016 CIE Conference on Radar, Explain the main difference
2- Please show the novelty in the work and the enhancement in probability of detection
3- From line 82 to 86 needs to rewrite appropriately (repeated sentences)
4-In sec 3, you didn't explain any impact of "Number of Receiving Antenna Elements"
5-How did you overcome the phase shift Resulting by the distance between the three antenna
6- In figure 6 the 'false alarm probability' with three antenna is higher of the single element , please explain
Reviewer 2 Report
The experimental data should be used to verify the effectiveness of the proposed method.
Reviewer 3 Report
1. The authors assume that diffuse multipath is negligible compared with specular multipath for this detection problem, and they cite Barton [6] and Blair [7] to support this assumption. However, both Barton and Blair show that diffuse multipath is in fact significant under certain conditions. This would depend on sea state, grazing angle, range of the target, wind speed, wind direction, polarization, radar frequency, range resolution, etc. Therefore, the authors should consider modelling both diffuse and specular multipath in future work.
2. The authors have chosen the test statistic for detection as the maximum of the absolute values of the 3 differences between the 3 receive antennas. The authors showed interesting plots to explain why they decided to use this test statistic. However, this test statistic is not optimal for detection. For future work, the authors should derive the optimal test statistic for detection.
Reviewer 4 Report
A radar detection method that uses a single transmitting antenna and three receiving antennas was proposed in this study. Comparative and simulated experiments confirmed its effectiveness. However, the introduction needs to be revised. Paper content should be enriched. The language also needs further revision. These issues must be addressed and require major revisions.
1. Although as a communication type of article, the article is still relatively short.
2. Moderate English changes required. Reduce the expression of “first person”.
3. Introduction is not reach enough. Some key papers about Radar signal processing should be discussed:
a) Automatic pixel-level detection of vertical cracks in asphalt pavement based on GPR investigation and improved mask R-CNN, https://doi.org/10.1016/j.autcon.2022.104689
b) Radar Maneuvering Target Detection Based on Product Scale Zoom Discrete Chirp Fourier Transform, https://doi.org/10.3390/rs15071792
4. Figure 3, why only 45 km was used to analyze the relationship between radar antenna height and target height?
5. In figure 7, the theoretical and simulated values seems to be equal, what is the error between them?
6. Figure 7 and 8 are important, however, more deeply analysis should be provided.
7. Conclusion: author should state the limitation of this study.
Round 2
Reviewer 1 Report
The author answered on all comments and The paper can be accepted
Author Response
Thanks to reviewer 1 for the acception recommendation.
Reviewer 2 Report
The method presented in this paper is classic. There isn't any innovation in this paper.
Experiment is needed to very the method.
Author Response
Thanks to the reviewer 2 for the valuable suggestion. We agree with the reviewer 2 that the method proposed in this paper had better be verified by outside experiment. Due to the lack of the experimental data at present, we have to verify the method by simulation. We will try our best to implement the outside experiment and verify the method by real measured data.
Thank you again.
Reviewer 4 Report
The author has done a good job with the revisions. I have no further questions/remarks.
Author Response
Thanks to the reviewer 4 for your comments.